



**A Portable Nitrogen Dioxide Instrument Using Cavity-Enhanced Absorption**
**Spectroscopy**
Steven A. Bailey[1] , Reem A. Hannun[2], Andrew K. Swanson[1,3],  Thomas F. Hanisco[1]
1.  Atmospheric Chemistry and Dynamics Lab, NASA Goddard Spaceflight Center,

6          Greenbelt, MD, USA

2.  Atmospheric Science Branch, NASA Ames Research Center, Moffett Field, CA
3.  SciGlob Instruments and Services, LLC, Columbia, MD, USA
**Correspondence:** Steven A. Bailey (steven.a.bailey@nasa.gov)
**Abstract**
The Portable (2.7 kg) Cavity-enhanced Absorption of Nitrogen Dioxide (PCAND) instrument for
measuring *in situ* nitrogen dioxide ($NO_2$) was developed using incoherent, broadband cavity-
enhanced absorption spectroscopy (IBBCEAS). An LED light source centered at 408 nm was
coupled to a cavity 15 cm in length, achieving an effective optical pathlength of ~520 m. Our
precision was measured as 94 pptv (1 s). To date, we have flown this instrument on 3 balloon and
1 UAV test flights. This instrument records data to an SD card and outputs data (via an RS232
port) to external devices including a commercial radiosonde (iMet) for real-time data downlink.
**1    Introduction**
Nitrogen dioxide ($NO_2$) is a major contributor to air pollution in the Earth's troposphere. Its main
source is a byproduct of combustion from the burning of fossil fuels (Spinei, E. *et al.* 2014). $NO_2$
has been monitored from satellite instruments (like OMI, TROPOMI, and GEMS) for a decade
(Miyazaki, K. *et al.* 2017, Duncan, B. *et al.* 2015, Martin, R.V. *et al.* 2003, Cooper, M.J. *et al.* 2020),
providing a global understanding of emissions and air quality. However, satellite retrievals of the
total column $NO_2$ rely on estimates of the vertical distribution of $NO_2$ based on models or
climatology (Cersosimo, A. *et al*, 2020).  These *a priori* estimates are a major source of uncertainty



in making retrievals of $NO_2$ columns from satellite measurements (Cooper, M.J. *et al*. 2020, Dang,
R. *et al*. 2023).

Direct measurement of the vertical profile can verify and improve these *a priori* estimates.
Aircraft instruments cannot typically make a continuous vertical profile of the atmospheric
column. Therefore, an instrument small enough to fly on a balloon (or drone) is needed.
Techniques for measuring in situ $NO_2$ include Laser Induced Fluorescence (LIF)(Thornton, J.A. *et*
*al.* 2000), various-optical, absorption methods (like IBBCEAS)(Womack, C.C. *et al*. 2022, Min K.E.
*et al*. 2016), and chemical techniques (like chemiluminescence)(Ryerson, T.B. *et al.* 2000).
Although all these techniques have their pros and cons for use, we chose to focus on optical,
absorption methods for several reasons. First, we have successful experience using IBBCEAS in a
previous, ozone ($O_3$) based instrument (Hannun, *et al.*, 2020). Second, stability and ease of
calibration are desirable, which we found to be the case with the $O_3$ instrument. Third, the
technique can be scaled to a small enough size and weight to fly (via balloon or drone) into the
free troposphere. An instrument using LIF to measure $NO_2$ would (in our experience) not be
suitable for our purposes. Its size and weight would be too great to work with a small weather
balloon, despite LIF having greater sensitivity than IBBCEAS. Previously, a small $NO_2$ instrument
was developed by the Royal Netherlands Meteorological Institute (Dutch: Koninklijk Nederlands
Meteorologisch Instituut, KNMI) (Sluis, et al., 2010). That instrument uses chemiluminescence to
measure $NO_2$, with a reported precision of 7.7 ppbv/sec. Although chemiluminescence
instruments fit our size and weight criteria, they suffer from a lengthy calibration procedure
before every flight. Additionally, an instrument using chemiluminescence does not have the
desired sensitivity we require.

A description of our instrument using IBBCEAS follows. Performance metrics will show our
instrument meets the Federal Aviation Administrations (FAA) uncontrolled, maximum allowable
weight (~2.7 kg) for a balloon payload. In addition, we record sensitivity to $NO_2$ that exceeds the
KNMI sonde by more than an order of magnitude. A description of our calibration procedure is
detailed showing its simplicity. Finally, we demonstrate an atmospheric vertical profile



measurement from one of our balloon flights. We also validate our instrument via a ground-
based comparison with another established $NO_2$ instrument.
**2    Principle of operation**
IBBCEAS is an established technique for the detection of trace gases (Fiedler *et al.*, 2003; Ball *et*
*al.*, 2004; Washenfelder *et al.*, 2008) including $NO_2$ (Min *et al.*, 2016). We use an LED as our
incoherent, broadband light source centered at 408 nm. This is coupled to an optical cavity with
highly reflective mirrors on either end. IBBCEAS leverages the mirror reflectivity to turn a
physically short path length (15 cm) of the cavity into an effective optical pathlength of ~520 m.
This effective pathlength increases the probability of $NO_2$ absorption in the cavity, thereby
increasing the sensitivity (94 pptv @ 1 s) of the instrument.

Shown in Figure 1, output from an LED is collimated into the gas sample cell (cavity) where it first
passes through the leftmost mirror. Both mirrors have highly reflective coatings (>99.9%) on
curved surfaces (r=250 mm) facing towards each other. Only a small fraction of light enters the
cell, but the light (photons) bounces back and forth between both mirrors thousands of times on
average before exiting the rightmost mirror. Photons that exit are then detected by a silicon
photomultiplier (SiPM). A transconductance amplifier is then used to convert small amounts of
current from the SiPM into measurable voltage levels. A micro controller with a 12-bit analog to
digital convertor digitizes this voltage. The micro controller is both a data acquisition system and
a controller of the LED and 3-way valve. A digital lock-in scheme is used to remove background
light by modulating the LED at 100 Hz with a large duty cycle (70%).

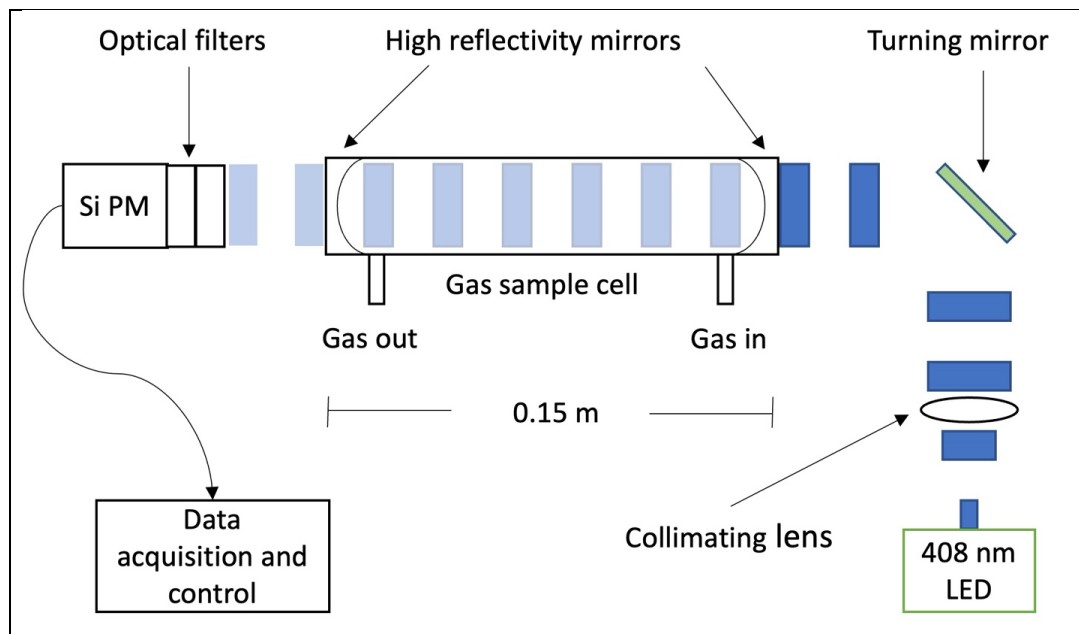

**Figure 1.** Incoherent broadband cavity enhanced detection technique for $NO_2$. A LED at 408 nm is collimated and coupled into the detection cell via high reflectivity mirrors ($R > 99.98\%$), creating a long optical pathlength. The light attenuated by the sample is then detected using a silicon photomultiplier (SiPM).


Trace gas absorption (using IBBCEAS) is a measurement of light attenuation. As light is absorbed
and scattered (via Rayleigh), an attenuation of light is seen at the SiPM. The Beer-Lambert
absorption coefficient, $\alpha_{abs}$, is directly related to the light intensity exiting the cavity
(Washenfelder *et al.*, 2008; Hannun *et al.*, 2020) through the equations:

$$\alpha_{abs} = \left(\frac{I_0 - I}{I}\right)\left(\frac{1-R}{d} + \alpha_{Ray}\right) \qquad (1)$$
$$\alpha_{cav} = \left(\frac{1-R}{d}\right) \qquad (2)$$
$$L_{\text{eff}} = \left(\frac{1}{\alpha_{cav}}\right) \qquad (3)$$



Here $I_0$ is the intensity of light in the absence of any absorbing molecules, $I$ is the intensity of
light including absorbing molecules, $R$ is the mirror reflectivity, $d$ is the physical distance between
cavity mirrors, and $\alpha_{Ray}$ is the extinction due to Rayleigh scatter. The term $(1-R)/d$ is the
theoretical cavity loss, $\alpha_{cav}$. $L_{\text{eff}}$ represents the maximum effective pathlength. In the case of
mirrors with R=0.9997, the maximum theoretical $L_{\text{eff}}$ for our 15 cm cell would be 450 m.
**3    Instrument description**
PCAND is housed in a small aluminum box measuring 38 cm length x 22 cm width x 7 cm height
with a total weight of 2.7 kg. Inside the box (Figure 2) is an optical plate where all the instrument
components are mounted. Power comes from an 11.1 volt Lithium Ion rechargeable battery with
2200 mAh (24 Wh) of storage. Table 1 summarizes the PCAND design and performance
characteristics.

Table 1. Summary of PCAND performance capabilities

| Specification | Value |
|---|---|
| Size | 38 x 22 x 7 cm |
| Weight | 2.7 kg |
| Power | < 6 W |
| Data rate | 1 Hz |
| Precision ($1\sigma$, 1Hz) | 2.3 x $10^9$ molec. $cm^{-3}$ |
| Accuracy | 6.0% |
| Time response | 3 s |




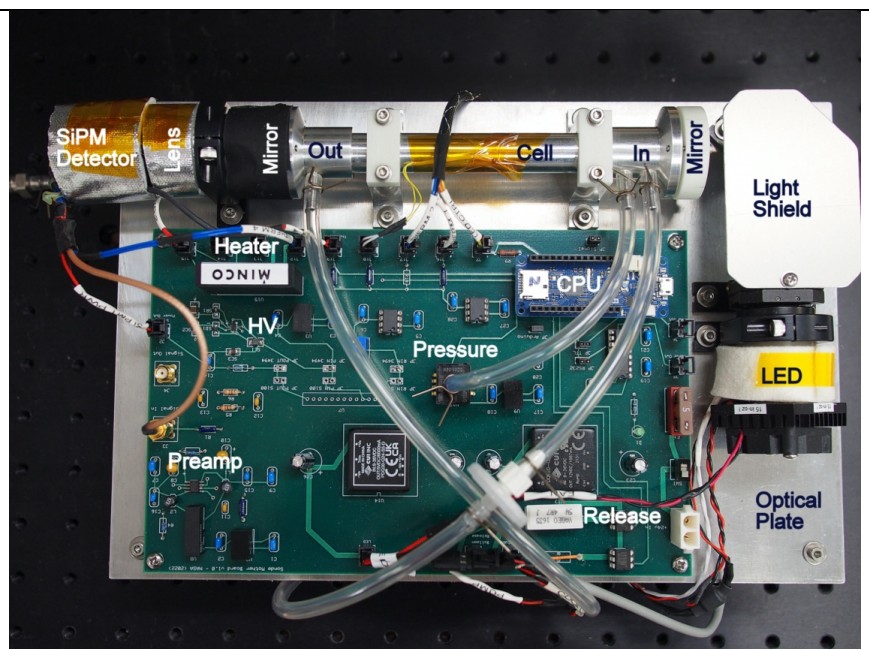

**Figure 2.** A top view of the NO$_2$ instrument. Major components include A) the optical plate, which consists of the LED assembly, light shield, turning mirror (under light shield), the optical cell, end mirrors, collimating lens, and SiPM detector; B) The electronics motherboard with detector preamp, heater controller, pressure sensor, balloon release circuit, and the data acquisition system (CPU). Not shown is the nafion tubing used to dry the air before entering the instrument.


## 3.1 Optical components

### 3.1.1 LED assembly

A UV LED ($\lambda_{max}$ = 408 nm, FWHM = 30 nm) (Thorlabs M310D1) is mounted to a custom heat sink and temperature controlled to 25 °C with a thermo-electric cooler controller (Thorlabs MTD415T). Constant current to the LED is supplied by a low noise controller (Thorlabs MLD203CLN). The LED assembly includes a 15 mm focal length collimating lens (Thorlabs LA1074-A) followed by a turning mirror (Thorlabs PF10-03-F01) to direct light into the sample cell.

109



### 3.1.2 Sample cell

The sample cell is manufactured from an aluminum alloy tube measuring 15 cm in length with a 1.4 cm inner diameter. The cell mirrors (Layertec 103654) have a reflectivity of $R > 99.9\%$ over the detected spectral range (Figure 3) and a 250 mm radius of curvature. Mirrors are held to the cell ends with bezel mounts on flanges with face seal o-ring glands. Although the mounts themselves are non-adjustable, they are fabricated to hold the mirrors in a way that maximizes their centricity to the cell ends. Furthermore, the incoherent light source negates the need for rigid mirror alignment. A pressure transducer (Honeywell ASDXACX015PAAA5) measures the cell pressure from a port near the cell inlet.

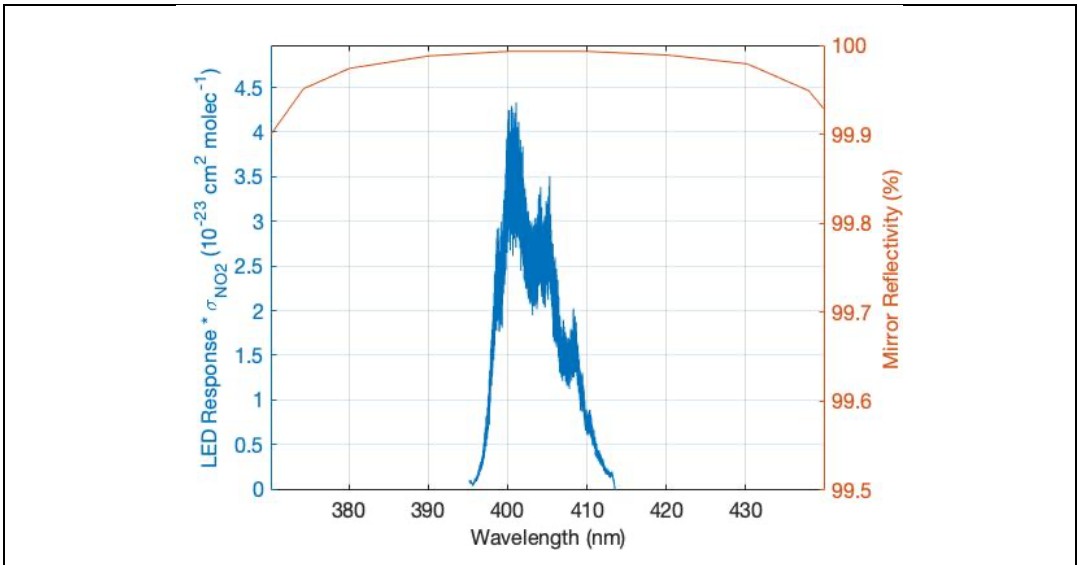

**Figure 3.** Normalized LED spectral response x NO$_2$ cross-section vs mirror reflectivity (99.98% @ 408 nm). The LED ($\lambda_{\max}$ = 408 nm, FWHM = 30 nm) response was measured using a grating spectrometer with the instrument SiPM and associated detector optics. The absorption cross-section of NO$_2$ (for this instrument) is the integration of the above product (with a resolution of 0.0005 nm) which yields 6.0419 x 10$^{-19}$ cm$^2$ /molecule.



*3.1.3    SiPM assembly*
Following exit from the sample cell, light enters an optical bandpass filter (Semrock FF01-405/10-
25), then a lens (Thorlabs LA1252-A) focuses the beam onto a Silicon Photo Multiplier (SiPM -
Onsemi 30035) detector. The detector is biased by ~29 volts DC via a LT3494A boost converter.
This voltage sets the gain of this device. Signal from the SiPM is amplified through a
transimpedance amplifier based on a low noise, ADA4625-2 op-amp. The SiPM assembly is
thermally stabilized by heating it to a 35 °C setpoint using a Minco CT335 heater controller. The
temperature of the SiPM is monitored with a 10K thermistor mounted adjacent to the heater.
Temperature of the detector is held to within 0.1 °C of the setpoint using the Minco controller.
**3.2    Flow system**
The PCAND instrument uses a small, 12 volt diaphragm pump (Parker E134-11-120) to achieve a
1.4 standard liters per minute (SLM) flow rate. Flush time is approximately 3 seconds as evident
from Figure 4. A 3-way valve (ASCO 411L3212HV) is used to switch the flow between sample air
and scrubbed air (via an inline charcoal filter). The charcoal filter removes any $NO_2$ from the flow
and gives us our $I_0$ (reference) measurement every 30 seconds for 5 seconds, leaving us sample
air measurements 50 seconds out of every minute.




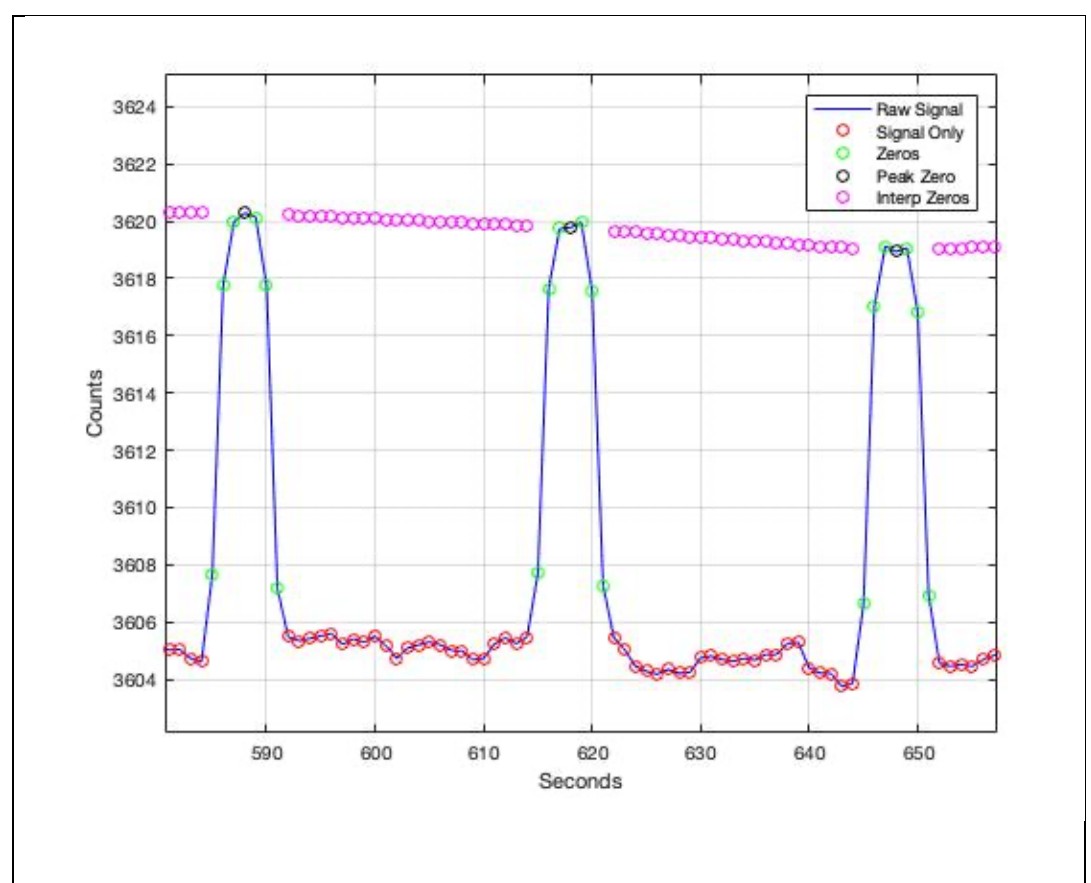

**Figure 4.** Cadence used to make real-time measurements of $I_z$ (signal with no absorbers) vs I (signal with absorbers) is 7 seconds for $I_z$ and 23 seconds for I. We found this was a good cadence allowing 3 seconds to achieve peak $I_z$ and 3 seconds to return to I. A charcoal filter is switched into the airflow to achieve the $I_z$ measurement.


We expect a small interference from water vapor. H2O vapor has a cross-section of $3 \times 10^{-17}$
$cm^2$ /molecule at 405 nm (Lampel *et al.*, 2015; 2017). An atmospheric abundance of $H_2O$ = 1%
contributes the same absorption as 50 pptv of $NO_2$.  In addition, we notice stronger attenuation
that is not consistent with gas phase absorption like that reported in ozone instruments using
UV absorption (Wilson *et al.*, 2006).  In principle the presence of water vapor should not affect
the measurement if the abundance is constant between the sample and the scrubbed air.
However, the scrubber material (activated charcoal) can add or remove water vapor to the
sampled air depending on the prior humidity. Because of this interference, water vapor is
removed using two 30 cm lengths of 0.3 cm diameter Nafion Dewline tubing held in an
enclosure with drierite. The dry air sample eliminates any contribution of water vapor in the
measurement.

PCAND uses fluorinated ethylene propylene (FEP) lined thermoplastic tubing for all internal
plumbing. A 2-micron teflon membrane filter is positioned immediately before the cell inlet to
keep small particles from entering the cell and potentially dirtying the mirrors.
**3.3    Data acquisition**
PCAND uses an Arduino MKR Zero microcontroller for 3-way valve control, LED modulation, and
data acquisition. Arduino actuation of the valve is made through a CoolCube R valve controller,
which reduces the holding current needed to keep the valve in its open state. LED modulation is
produced by the Arduino through the LED controller at a 100 Hz rate. This modulation has a 70%
duty cycle used to achieve a digital lock-in to remove any background light from the absorption
measurement. We oversample the absorption signal 42k samples / second to increase the native
Arduino internal 12-bit measurement to an effective (averaged over a second) ~21-bit
measurement. Data is recorded both to an SD card and sent to an RS-232 port. The latter is useful
for both instrument testing and for connecting to an external iMet radiosonde where the data is
merged for RF data downlink by the radiosonde.
**3.4    Data processing**
The PCAND absorbance calculation uses equation 1, but accounts for the differential cell pressure
between the sample flow and the zero flow, which is restricted by the scrubber. Including the
Rayleigh scattering for both zero air and sample air, Eq. 1 is rewritten as equation 4 (Min *et al.*
2016 ; Hannun *et al.*, 2020) :

$$\alpha_{NO2} = \left(\frac{I_Z}{I} - 1\right)\left(\alpha_{cav} + \alpha_{Ray,Z}\right) + \Delta\alpha_{Ray} \qquad (4)$$
$$\Delta\alpha_{Ray} = \alpha_{Ray,Z} - \alpha_{Ray,S} \qquad (5)$$





$$\alpha_{Ray} = N_{air}\sigma_{Ray} \tag{6}$$
$$\alpha_{NO2} = N_{NO2}\sigma_{NO2} \tag{7}$$
$$\alpha_{Ray,S} = \left(\frac{I_Z}{I} - 1\right)\alpha_{cav} \tag{8}$$

Zero air is NO$_2$ scrubbed air where $I_Z$ substitutes for I$_0$ (from equation 1). Rayleigh cavity
extinction is broken into 2 parts ($\alpha_{Ray,Z}$ and $\alpha_{Ray,S}$) describing zero air and sample air cavity
extinction respectively. In both cases, the Rayleigh scattering cross-section ($\sigma_{Ray}$), weighted by
the SiPMT response curve (Figure 3), is used (Bucholtz, 1995). The NO$_2$ number density
(concentration) is found by knowing the absorption cross-section of NO$_2$ ($\sigma_{NO2}$) (Vandalae, 1998).
## 4 Performance
### 4.1 Sensitivity and calibration
The PCAND effective pathlength of the optical cavity determines the instruments sensitivity to
NO$_2$. Highly reflective mirrors on either end of the cavity are statically mounted, so no adjustment
of their position is required. In practice, the alignment is stable over months of operation. After
the initial alignment, calibration is needed to determine the effective pathlength given the mirror
positions. We can use equation 4 with known quantities of NO$_2$ to determine the effective
pathlength (Figure 5a). We can also use Rayleigh scattering alone to solve for effective pathlength
(Figure 5b). This requires varying the pressure of zero air (in the absence of NO$_2$) to generate a
data set of absorption attenuation (*I*) vs number density of zero air. It also requires we solve for
equation 8 after it has been reduced from equation 4. To do this, we must assume $\alpha_{Ray,Z}$ is taken
at vacuum, so $\alpha_{Ray,Z}$ goes to zero leaving only $\alpha_{Ray,S}$ . We must calculate ($I_z$) at vacuum using
our data set. This leaves us with equation 8 to solve for effective pathlength (equation 3). Using
known quantities of NO$_2$ and equation 4 yields a pathlength of 519 ± 2 m. Using the Rayleigh
scattering method and equation 8 yields a pathlength of 524 ± 1 m. The two methods of
calibration are within < 1% of each other and both yield pathlengths that agree to within $2\sigma$
uncertainty for each fit.  Therefore, we choose to use the Rayleigh scattering method in future
calibrations (when needed) of PCAND. Note that due to the small Rayleigh cross-section of air at



408 nm, sigma = 1.5 x $10^{-26}$ cm$^{-2}$/molecule (Bucholz, 1995) the calibration using air is susceptible
to leaks and contamination. Adequate care must be taken to ensure that the system is free of
leaks and that the air is pure.

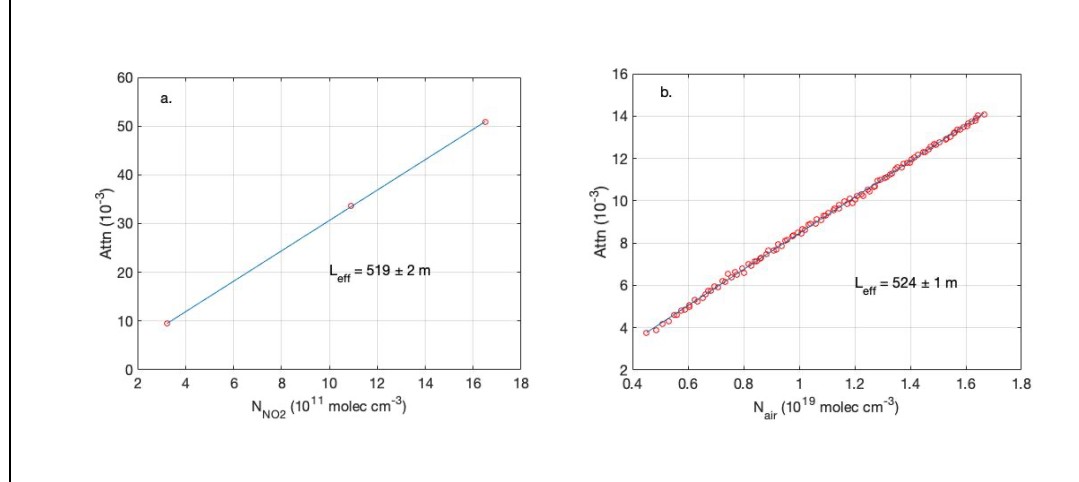

**Figure 5.** NO$_2$ Sonde calibration: a) The effective pathlength ($L_{eff}$) as determined by attenuation (Attn) due to known additions of NO$_2$ from a reference tank of NO$_2$ mixed with zero air. The slope yields the effective pathlength as determined from Equation 1 in the text using the known NO$_2$ absorption cross section; b) Attenuation due to Rayleigh scatter over a range of cell pressures. The slope of attenuation as a function of number density gives the pathlength using the known Rayleigh scattering cross-section for zero air. The pathlength from each calibration agreed to within $2\sigma$ uncertainty for each fit.


**4.2   Precision and accuracy**
The PCAND precision was determined by flowing zero air (under constant pressure of 920 mbar)
into the cavity for 2 hours while accumulating 1Hz data. Figure 6 is an Allan deviation plot showing
a 1 Hz precision of 94 pptv and a 10 s precision of 30 pptv. The 1 Hz precision translates to 2.3 x
$10^9$ molecules cm$^{-3}$ of NO$_2$ at 1 atmosphere.



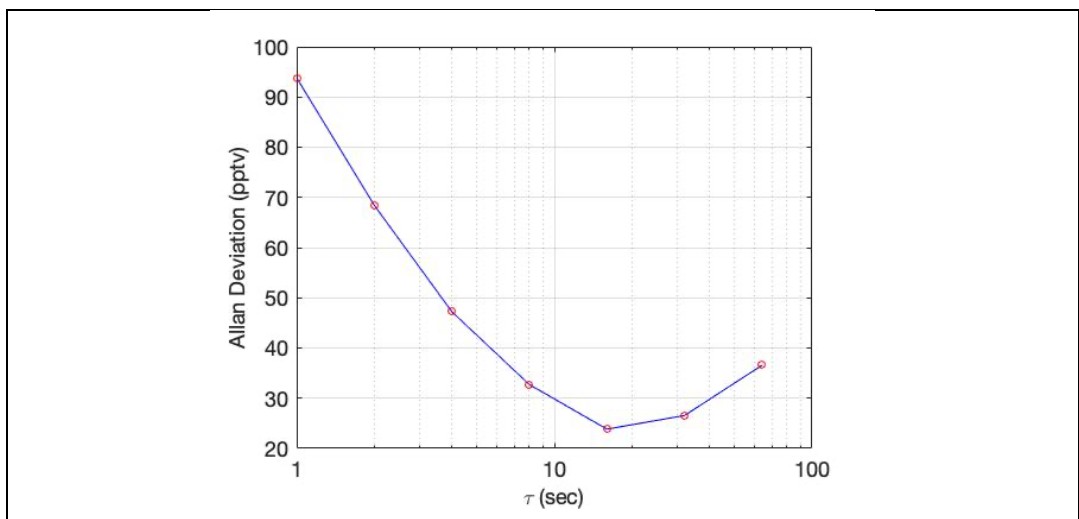

**Figure 6.** Allan deviation plot for 2 hr of sampling zero air at constant pressure (940 mbar). The Allan deviation is expressed in pptv equivalents of $NO_2$ as a function of the integration time $\tau$. The curve shows a precision of 94 pptv at 1 second integration time.


The accuracy of PCAND measurements depends on $NO_2$ and Rayleigh cross section uncertainties,
pressure sensor uncertainty, thermistor uncertainty, and cavity extinction uncertainty. The $NO_2$
absorption cross-section uncertainty is reported to be 3% (Spinei, 2014; Vandalae 1998). We use
3% for the Rayleigh scattering cross-section uncertainty (Bucholtz, 1995). From data sheets, we
conclude temperature and pressure measurements to have uncertainties of 0.5% and 2%
respectively. We also measured cavity extinction slope uncertainty at 1%. Together, the total
uncertainty when propagated through equation 4 comes to 6 % when applied to the final $NO_2$
number density.
**4.3    Response time**
Response time is a direct function of gas flush time in our cell given our small vacuum pump. A
flow rate of 1.4 SLM is achieved with our pump resulting in a response time of approximately 3
seconds (Figure 4). Given our cadence of 5 second zero followed by 25 second sample, we can
see (by eye) it takes ~3 seconds for the signal to stabilize with zero air. A larger pump could
shorten this response time at the expense of more mass and power needed.





**4.4    Photolysis Effects**
The photolysis quantum yield is 0.22 at 408 nm (*Troe, 2000)*, so we expect some fraction of the
$NO_2$ in the cell to photolyze, $NO_2 + h\nu \rightarrow NO + O$.  In static cells the photolysis of $NO_2$ has been
shown to be a concern (Platt *et al., 2019*) In the case of our detection, it is unlikely that a
significant fraction of $NO_2$ will be photolyzed because the sample flows through the cell quickly
with a flush time of approximately 1 s and the number of photons available for photolysis is
small.
We can estimate the number of photons in the cell from the detector signal. The SiPM has a
radiant sensitivity of $4 \times 10^5$ A/W and a photon detection efficiency of 50%.  Based on our
detection signal of $2 \times 10^{-5}$ A, we estimate the optical power is roughly $10^{-10}$ W and calculate a
photon flux of $2 \times 10^9$ photons/s.  A typical absorbance with 1 ppb $NO_2$ in the cell is $10^{-3}$, thus
we expect that roughly $2 \times 10^6$ photons/s are absorbed by the 1 ppb $NO_2$ in the cell. At 900 hPa
the number density of 1 ppb $NO_2$ is roughly $2.2 \times 10^{10}$ molecules/cm$^3$. The absorption of $2 \times 10^6$
photons would result in the photolysis of $4.4 \times 10^5$ $NO_2$ molecules, or $2 \times 10^{-5}$ of the available
$NO_2$ molecules.  While this number is quite low for our conditions, it is worth noting that with
slower flows and higher photon fluxes the photolysis could be significant and secondary
chemistry could be a concern.
**5    Field demonstration**
PCAND was launched on 3 low altitude (~7 km) balloon flights for demonstration purposes during
the summer of 2022. PCAND was physically linked (via RS232 cable) to a commercial weather
sonde for real-time data downlink (via the weather sonde). Results from the second flight (Figure
7) launched on 18 August 2022 show a vertical profile of $NO_2$ indicative of that time of year with
high concentrations of $NO_2$ near the ground. This flight occurred at 8 am local time when the
boundary layer was still close to the ground. The temperature deviation in the instrument box
during flight to 7km was less than 1 °C.



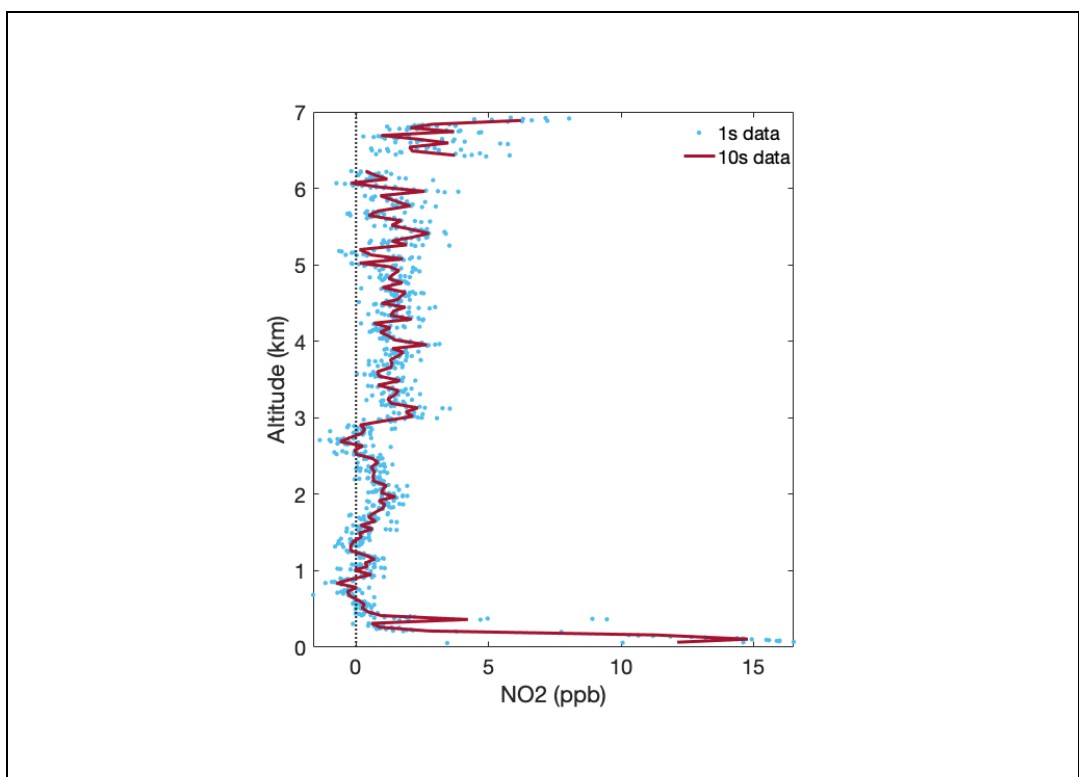

**Figure 7.** NO2 Sonde flight data from 18 August 2022 balloon launch. Programmed cut-down of balloon at 7 km to achieve payload recovery. Significant concentrations of NO2 appear near the surface and again at cut-down (~7 km) altitude.



## 5.1    Validation with CANOE


PCAND was validated with another NO$_2$ instrument called CANOE (Compact Airborne Nitrogen
diOxide Experiment). CANOE was based on the design of a similar instrument called CAFE (St.
Clair *et al.*, 2019) (Compact Airborne Formaldehyde Experiment). The only difference between
CANOE and CAFE are the laser wavelengths (532 nm for CAFE vs 355 nm for CANOE) and PMT
detectors used. CANOE is an LIF instrument which has been deployed on several airborne
campaigns including Dynamics and Chemistry of the Summer Stratosphere (DCOTSS) and Fire



Influence on Regional to Global Environments and Air Quality (FIREX-AQ). CANOE has been
calibrated to known cylinders of $NO_2$ concentration. Figure 8 shows a ~4-hour data set where
PCAND and CANOE shared the same inlet port sampling the air during a morning in the DC greater
metropolitan area. Clearly, a rush hour peak of $NO_2$ is seen trailing off by noon. Figure 8a shows
good agreement between the measurements with a slope of $0.94 \pm 0.004$ and an intercept of
$0.09 \pm 0.012$ ppbv $NO_2$ ($r^2$ = 0.96).

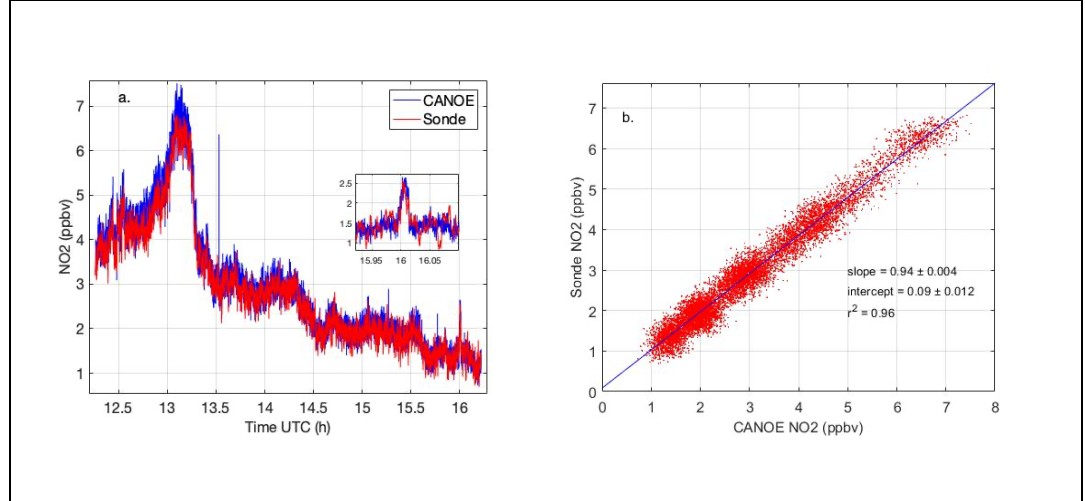

**Figure 8.** $NO_2$ Sonde and CANOE measurements during rush hour at GSFC on 14 July 2023. a)
Comparison over ~4 hours which clearly shows rush hour peak around ~13 UTC. b) Scatter
plot of the same data showing high correlation between instrument measurements. A linear
fit to the data gives a slope of $0.94 \pm 0.004$ and an intercept of $0.09 \pm 0.012$ ppbv with an $r^2$
= 0.96.

**6    Summary and conclusions**
PCAND provides very high sensitivity to $NO_2$ for such a small package using broadband cavity-
enhanced UV absorption at 408 nm. PCAND has a precision of ~94 pptv $s^{-1}$ with an accuracy of
6.0%.  Although PCAND was designed for portable, battery powered operation (as needed for a
sonde or drone flight), it could easily be used in either ground or lab-based measurements. It was
successfully tested on 3 balloon flights producing $NO_2$ vertical profiles for each. A comparison



with another (calibrated) NO$_2$ instrument (CANOE) showed strong agreement over a ~4-hour
period.
*Author contributions.* SAB performed the investigation, controller software, electronics design,
testing, and wrote the paper. RAH wrote the signal processing code and determined the best
wavelength to use for NO$_2$ absorption. AKS did all the mechanical design including optical plate,
fixtures, and cell. TFH determined the correct mirrors to use, consulted with AKS on the optical
layout, and made the science case for receiving funding for this work.

*Competing interest*. At least one of the (co-)authors is a member of the editorial board of
Atmospheric Measurement Techniques.

*Acknowledgements*. The balloon flights were made at the Howard University Beltsville Campus
with the help of Adrian Flores. The drone flight was made at Virginia Commonwealth University
Rice Rivers Center with the help of Gregory Garman and Ron Lopez. We would additionally like
to thank Ryan Stauffer for his expertise in balloon flight needed to launch and recover our
instrument.

*Financial support*. This research has been supported by the NASA Internal Research and
Development (IRAD) program at Goddard Space Flight Center (GSFC).

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
