# Peer review of "A Portable Nitrogen Dioxide Instrument Using Cavity-Enhanced Absorption"

_Atmospheric Measurement Techniques, 2024_

## Author Response (AR1)

1) Comment from reviewer #2: *LED spectrum stability: LED emission spectrum can be very sensitive to its temperature and as you described, the LED here is temperature controlled at 25 degrees. Can you show that this is also the case at high altitudes with environment temperatures well below zero? presumably, that would make life easier for the TEC unless the LED is not generating enough heat.*

   Response: Thank you for your comment. Our instrument was designed to fly on a drone up to 400 feet or on a balloon in the boundary layer. Given flight data shown in Figure 7 goes to 7km before balloon cutdown, our temperature in the instrument enclosure (which is insulated) deviated less than 1 deg C between launch (at ~2350 seconds) and cutdown (at ~3600 seconds) as seen in the plot below. I added a sentence to the current paper (line 249) indicating this temperature deviation.

[Figure]

2) Comment from reviewer #2: *Comment optical alignment: I can see how fine alignment of the cell is not required. However, the device can experience a significant temperature gradient from ground level to a height of several km. In the altitude profile you presented this gradient can easily reach 40-50 degrees. With thermal expansion, especially when part of the device is kept at 25 C, how confident are you that the device is not too misaligned? can you provide*

*data similar to Figure 4 from an actual flight? with temperature profile and LED temperature if you can.*

Response: Thank you for your comment. As I stated in my response to comment #1, our instrument is contained in an insulated enclosure with thermal controls for both the detector and LED. Given that and the small temperature deviation in the enclosure during flights through the boundary layer (< 1 deg C), we are not concerned with atmospheric temperature effects for these intended flights.

3) Comment from reviewer #2: *section 3.2 is confusing to me. presumably, you are referring to the molecular absorption cross-section which should have cm2 or cm2 / molecule units. you used molecules / cm2. For water vapour, this should be orders of magnitude lower.*

Response: Thank you for your comment. You caught two typos, one of which I caught myself and corrected in the current document which now uses the cross-section units of cm2 / molecule for water vapor. However, I did not catch the lower water vapor number. Thank you again. I will correct that number to $3 \times 10^{-27}$ in the next version of paper.

4) Comment from reviewer #2: *in the caption of Figure 8, you refer to the device as a "Sonde" although it was not flown here.*

Response: Thank you for your comment. I will remove the words "NO$_2$ Sonde" and replaced them with the word "PCAND" in the caption of Figure 8 for the next version of the paper.

5) Comment from reviewer #2: *alongside a comparison to a state-of-the-art device at ground level I would like to see a comparison while performing under the intended instrument usage - i.e. during flight, under severe temperature and pressure gradients, and atmosphere composition changes.*

Response: Thank you for your comment. Although we would very much like to compare our instrument (in flight) with "a state-of-art device", there does exist such a device that is small enough to fly on a weather balloon or drone that measures NO2. Regarding our instrument, we believe we have already flown it "under the intended instrument usage".

6) Comment from reviewer #2: *under Acknowledgements, you refer to drone flights. did I miss that in the main text? I didn't see a reference to any drone flights.*

Response: Thank you for your comment. Although we did have a drone flight, you are correct in that I do not mention it in the paper. Thank you for catching that. I will remove the sentence about drone flight in the Acknowledgements.

7) Comment from reviewer #2: *I would like to see a discussion about the possible photolysis of NO2 at 405 nm considering its high quantum yield, or at least an explanation of why it is perhaps insignificant in this case.*

Response: Thank you for your comment. I added a new section 4.4 (Photolysis Effects) to the current paper (line 226) that addresses this comment. Here is that text:

**Photolysis Effects**

The photolysis quantum yield is 0.22 at 408 nm (*Troe, 2000)*, so we expect some fraction of the $NO_2$ in the cell to photolyze, $NO_2 + h\nu \rightarrow NO + O$. In static cells the photolysis of $NO_2$ has been shown to be a concern (Platt *et al., 2019*) In the case of our detection, it is unlikely that a significant fraction of $NO_2$ will be photolyzed because the sample flows through the cell quickly with a flush time of approximately 1 s and the number of photons available for photolysis is small.

We can estimate the number of photons in the cell from the detector signal. The SiPM has a radiant sensitivity of $4 \times 10^5$ A/W and a photon detection efficiency of 50%. Based on our detection signal of $2 \times 10^{-5}$ A, we estimate the optical power is roughly $10^{-10}$ W and calculate a photon flux of $2 \times 10^9$ photons/s. A typical absorbance with 1 ppb $NO_2$ in the cell is $10^{-3}$, thus we expect that roughly $2 \times 10^6$ photons/s are absorbed by the 1 ppb $NO_2$ in the cell. At 900 hPa the number density of 1 ppb $NO_2$ is roughly $2.2 \times 10^{10}$ molecules/cm$^3$. The absorption of $2 \times 10^6$ photons would result in the photolysis of $4.4 \times 10^5$ $NO_2$ molecules, or $2 \times 10^{-5}$ of the available $NO_2$ molecules. While this number is quite low for our conditions, it is worth noting that with slower flows and higher photon fluxes the photolysis could be significant and secondary chemistry could be a concern.

1) Comment from Referee #3: *Line 16: 3 balloon and 1 UAV flight are listed in the abstract. At the end of the Introduction (line 56) it is stated that one flight is used. Please standardize across the paper.*

   Response: Thank you for your comment. I have removed all reference to the UAV flight in the paper. Although we did have 3 balloon flights, only one gave us useful $NO_2$ data as we were working out radio interference problems on the other two flights which contaminated the $NO_2$ data. I changed text in the **Summary and conclusions** section, sentence 4 (line 274), from "It was successfully tested on 3 balloon flights producing $NO_2$ vertical profiles for each." to "It was successfully tested on 3 balloon flights".

2) Comment from Referee #3: *Line 26: Check reference style for Cersosima et al. as shown in paper text with first initial.*

   Response: Thank you for your comment. I have corrected the reference on line 26 from a Calibri font to a Calibri (Body) font. Thank you for catching that.

3) Comment from Referee #3: *Line 34: remove comma between optical and absorption.*

   Response: Thank you for your comment. I have removed the comma from line 34 between optical and absorption.

4) Comment from Referee #3: *Lines 33-35: Check formatting of parentheticals with references. Also, list is missing any reference to electrochemical sensors.*

   Response: Thank you for your comment. I cleaned up the sentences in lines 33-35 to make the parentheticals clearer and more concise. I also just rewrote the **Introduction** (paragraph 2, line 30) to explain why we think existing electrochemical $NO_2$ sensors were not considered. The short answer is they do not have the precision and accuracy needed for determining a vertical profile of $NO_2$ from a balloon flight.

5) Comment from Referee #3: *Line 154: aerosol can also attenuate the light in the cavity, not just make the mirrors dirty.*

   Response: Thank you for your comment. It is true an aerosol can attenuate light, but more important is if said aerosol sticks to one (or both) of the mirrors. That effect on mirror reflectivity is profound and permanent until the mirrors are removed and cleaned.

6) Comment from Referee #3: *Line 178: $I_0$ font consistency (should be italicized).*

   Response: Thank you for your comment. I have gone through the paper and made all $I_0$ symbols italicized. Thank you for catching that.

7) Comment from Referee #3: *Line 221: Avoid personal pronouns. Replace "our pump" with "the pump", same on line 220.*

Response: Thank you for your comment. I have removed the personal pronouns for line 221 and 220 along with a few others I found in the paper. Thank you for catching those.

*I think this paper is suitable for publication, but the authors should address the following concerns first:*

1. *The authors compare PCAND to a recent lightweight LIF instrument, but should also compare its performance to other drone-based IBBCEAS NO2 instruments such as Zheng et al., 2024 and Womack et al., 2022.*

   Response: Thank you for your comment. PCAND was specifically designed for balloon flight. Therefore, any mention of Drone flights has been removed from the paper. Although we validated our instrument with a ground-based instrument (CANOE), our intent was to build an instrument light enough (less than or equal to 6 lbs) for balloon flight. We feel comparing our instrument to heavier airborne instruments is not applicable.

2. *The sample cell is made of aluminum alloy, but the authors later say that FEP tubing was used in all plumbing, presumably to reduce NO2 losses. Have the authors tested losses of $NO_2$ on the aluminum alloy surface? Similarly, did the authors test for $NO_2$ losses on the Nafion dryer? And is the charcoal filter expected to completely scrub out the NO2 or will there be a small fraction remaining?*

   Response: Testing for the loss of NO2 on surfaces was performed with NO2 concentrations typically found in the PBL, that is between 0.1 ppb – 20 ppb. Losses on the tubing between the inlet and cell were found to be most significant depending on the material, perhaps due to the large surface area to volume ratio in the 0.3 cm ID tubing. Teflon lined tubing was found to eliminate loss of NO2 within the detection limit of the instrument (< 0.1 ppb). Losses on the three-way solenoid valve, detection cell and the Nafion tubing were not significant within the detection limit of the instrument. The charcoal filter removes all of the NO2 in the 0.1 - 20 ppb range.

   We have modified the text in lines 164-168 (of the revised paper) to include this point.

3. *There are inconsistencies in how the mirror reflectivity is reported. Line 68 says >99.9%, the Figure 1 says >99.98%, and Line 91 says 99.97%, which correspond to significantly different values when converted to effective pathlengths.*

   Response: Thank you for your comment. All references to mirror reflectivity have been changed to R = 99.97%.

4.  *The authors should discuss the 3 second flush time in the context of the speed of the drone or balloon. What kind of vertical resolution will be expected with this smearing? Is it sufficient for atmospheric chemistry studies?*

    Response: Thank you for your comment. We don't expect any smearing during the 3 second flush time as this data is thrown out.

5.  *How frequently is the effective pathlength measured? Even if the cavity alignment is stable over months, is there any concern that mirror cleanliness will degrade faster than that?*

    Response: Thank you for your comment. We have been very surprised at how clean the mirrors have been over months of both field testing and lab use. Particles of dust are what have a largest negative effect on mirror reflectivity. Use of a particle filter before the cell has mitigated this problem quite well.

6.  *Section 4.1 is somewhat confusingly written. It's not really clear how these two methods are derived from the equations. Has this method been used before?*

    Response: Thank you for your comment. Yes, this method has been used before on an instrument (ROZE) we helped develop. (Hannun, *et al*., https://doi.org/10.5194/amt-13-6877-2020, 2020).
    Equation 1 comes from Washenfelder *et al*., https://doi.org/10.5194/acp-8-7779-2008, 2008. Equation 4 comes from Min, *et al*., K.-E., https://doi.org/10.5194/amt-9-423-2016, 2016.

    See my response to comment 7 below for clarification on equation 8.

7.  *Additionally, I would recommend moving equation 8 to section 4.1, because it doesn't follow from equation 7, but is rather derived in section 4.1*

    Response: Thank you for your comment. The leap to equation 8 is vague, so I moved it below the next paragraph with the following text added:

    By varying the pressure of the cell with zero air, we can extrapolate a value for $I_0$. Substituting $I_0$ for $I_Z$ in equation 4, we arrive at equation 8. At vacuum ($I_0$), both $\alpha_{Ray,Z}$ terms go to zero. The $\alpha_{NO2}$ term also goes to zero with no $NO_2$ in zero air.

$$\alpha_{Ray,S} = \left(\frac{I_0}{I} - 1\right)\alpha_{cav} \hspace{3cm} (8)$$

8. *Line 196: How are the "known" $NO_2$ concentrations provided? More detail is needed here.*

   Response: Thank you for your comment. Text from figure 5 "PCAND calibration: a) The effective pathlength ($L_{eff}$) as determined by attenuation (Attn) due to known additions of $NO_2$ from a reference tank of $NO_2$ mixed with zero air".

9. *Line 201: More details should be included about how leaks and contamination could affect the data. How would they affect the data? Are leaks independently checked for?*

   Response: Leaks or contamination can affect the calibration. The Rayleigh cross section of Air is small, sigma = 1.5 x $10^{-26}$ cm$^{-2}$ /molecule. Small amounts of strong absorbers can bias the calibration.  For example, the cross section of $NO_2$ is 6 x $10^{-19}$ cm$^{-2}$/molecule.  Adding 1 ppb of $NO_2$ to the air during a calibration results in a $(1x10^{-9})(6x10^{-19})/1.5x10^{-26}$ = 0.04 bias. Because leaks are pressure-dependent, in practice, a leak usually results in curvature of the Rayleigh calibration curve. We have added this statement at line 223-224 (of revised paper).

   Leaks could affect the measurement if the leaked air is different than the sampled air. In this case it depends on how big the leak is.  Yes, leaks are checked for, but this depends on the operator not the instrument.

10. *Figure 7: There are quite a few data points in this vertical profile with values close to -1 ppb. However, the reported uncertainty in the laboratory is 0.1 ppb. Do the authors expect that the precision degrades at higher altitudes? If not, how do they explain these negative values?*

    Response: Thank you for your comment. We experienced RF noise from the attached iMet weather sonde for data downlink. This was the best flight where the noise was partially mitigated. This accounts for the negative values. We since have moved the instrument to a thin, aluminum box which acts a faraday cage keeping all RF noise out.

11. *Additionally, the profile shows $NO_2$ concentrations of >5 ppb at 7 km, which is unusually high. Did this occur in all the profiles? Was it possible the flight was affected by lofted biomass burning plumes? The authors should discuss this in detail,*

*as accuracy at high altitudes will be critical if this instrument is to be used on balloon platforms.*

Response: Thank you for your comment. We can only speculate as to why the NO2 concentration was so high at 7 km. It could be a lofted biomass burning plume, but without an adjoining measurement from another instrument, it is impossible to say.